# *Orientia tsutsugamushi* Infection in Wild Small Mammals in Western Yunnan Province, China

**DOI:** 10.3390/pathogens12010128

**Published:** 2023-01-12

**Authors:** Yun-Yan Luo, Si-Tong Liu, Qi-Nan He, Ru-Dan Hong, Jun-Jie Zhu, Zhi-Qiong Ai, Jia-Xiang Yin

**Affiliations:** School of Public Health, Dali University, Dali City 671000, China

**Keywords:** *Orientia tsutsugamushi*, genotype, wild small mammals, the western Yunnan

## Abstract

Small mammals can transmit and serve as a reservoir for *Orientia tsutsugamushi* (*Ot*) in nature by carrying infected mites. In Yunnan, one of China’s main foci of scrub typhus, etiological evidence and genetic diversity for *Ot* is limited. A total of 2538 small mammals were captured seasonally from 2015 to 2016 in the three counties of Yunnan, and the spleen or liver tissue was examined for *Ot* based on 56 kDa nPCR. The overall prevalence of *Ot* was 1.77%, ranging from 0.26 to 9.09% across different species. The *Gilliam* strain was found in 35.6% (16/45) of the wild small mammals, followed by the *Karp* 11.1% (5/45) and *TA763* (1/45) strains, the last of which was discovered in western Yunnan for the first time. In Lianghe, *Ot* infection rates in wild small mammals were higher than in the other two counties. The infection rates of *Eothenomys miletus* with *Ot* were highest in the three dominant species. *Ot* infection rates in wild small mammals were higher in Lianghe (1200–1400 m) and Yulong (2800–3000 m). These findings could provide research clues for further confirmation of scrub typhus foci in western Yunnan or other similar natural environments.

## 1. Introduction

Scrub typhus caused by *Orientia tsutsugamushi* (*Ot*) is an emerging and remerging vector-borne disease. The disease has long been thought to be confined to the traditional distribution areas (East and Western central Asia to northern Australia), but it has now been reported in South America (Chile) and United Arab Emirates (Dubai). A novel *Orientia* species was found in Chile in 2016 [1,2,3,4], suggesting that the disease is more widespread than previously thought, and the global health burden of the vector-borne disease may currently be underestimated. In China, scrub typhus was first reported in 1948, and it has been recorded as a common infectious disease by the Chinese National Notifiable Infectious Disease Reporting Information System since 2006 [5,6]. In 2016, 22,558 cases were reported in China, 15.41 times more than in 2006 (1375 cases), translating to over 2 cases per 100,000 people. Yunnan province is one of the predominant endemic areas in China, the incidence of scrub typhus has increased enormously from 0.5 cases per million in 2006 to 6 cases per million in 2017, as well as approximately 50% (11593/21501) of the cases recorded in western Yunnan. Scrub typhus shows a reoccurrence in China, especially in Yunnan province. Therefore, there should be more research on the epidemic characteristics and transmission mechanism of the known endemic area of scrub typhus as a scientific reference for potential foci of similar environments.

Small mammals are the primary hosts for parasitic mites, and they play an important role in the transmission of *Ot* in nature because they can transport the infected mites from endemic areas to non-endemic areas. Humans are the occasional host when bitten by infected mites [7,8,9]. The different distribution and abundance of small mammals in nature determine the possibility for mites to parasitize and cause infection with *Ot*. In Korea, *Apodemus agrarius* was the most common species of small mammal infected with *Ot* [10]. In Thailand, *Rattus. rattus*, *R. losea*, and *Bandicota indica* were the dominant species [11]. In Vietnam, *R. norvegicus* were reported as the most commonly infected with *Ot* [12]. In China, *R. tanezumi*, *R. losea*, *Niviventer confucianus*, and *N. fulvescens* are represented as the dominant reservoir host of *Ot* in the south area of the Yangtze River. However, in the north area of the Yangtze River, *A. agrarius*, *A. peninsulae*, and *R. norvegicus* were the dominant species that carried *Ot* [13,14]. The density and distribution of mites have a direct influence on the incidence on scrub typhus [8]. Small mammals are the primary food source for the parasitic stage of mites, so the species and distribution of small mammals have an indirect influence on the spread of scrub typhus to humans and animals by influencing mite survival. Therefore, it is important to investigate the species and density of small mammals in areas with a high incidence of scrub typhus, contributing to our understanding of the spread of scrub typhus in nature and providing scientific clues for predicting the risk of scrub typhus to humans.

Studies on the infection of *Ot* among small mammals in China are mostly concentrated in North and Central China [13,14,15]. Yunnan province in Southwestern China (latitude, 21°8′ N to 29°15′ N, longitude, 97°31′ E to 106°11′ E) has diverse landscapes and natural conditions. It has a high elevation in the north and a low in the south, spanning eight latitudes, with a distinct vertical climate that includes cold, temperate, and tropical, providing an ideal environment for small mammals and mites to live and reproduce and for *Ot* to be maintained and transmitted in nature. Once the infection status of small mammals and infected mites spill over, there is an increased risk of human infection. However, only limited studies showed *R. tanezumi*, *Apodemus chevrieri*, *Eothenomys miletus*, *Niviventer andersoni*, and *N. confucianus* carried *Ot* in some areas of Yunnan province [16,17,18], and studies on human or wild small mammals in western Yunnan are still uncertain. Moreover, although scrub typhus in Yunnan has a long history, understanding of the genotypes of *Ot* in small mammals or humans is limited. Investigating *Ot* infection in small mammals can provide effective information for studying the transmission mechanism of *Ot* in the natural environment, as well as basic scientific information for further research into the high occurrence mechanism of scrub typhus in western Yunnan and other endemic areas. In this study, we aim to (1) detect the *Ot* infection in wild small mammals and characterize their genotypes in the three counties of western Yunnan and (2) explore the associated environmental factors of infection of *Ot* in wild small mammals in western Yunnan province.

## 2. Materials and Methods

### 2.1. Study Setting and Sampling Period

This cross-sectional study was conducted in western Yunnan province from 2015 to 2016, located in Lianghe, Yulong, and Jianchuan counties, respectively (Figure 1, created by R software (4.0.2 version)). Lianghe county is located in the southwestern portion of the Hengduan mountain, characterized by a south subtropical monsoon climate with rich water resources and lush vegetation. In the past decades, scrub typhus has shown a slowly increasing trend in Lianghe county, and the incidence peaked in 2017 at 98.58 per 100,000. Jianchuan and Yulong counties are located in the middle of the Hengduan mountain but are separated by low-lying valleys. Jianchuan has a low-latitude highland monsoon climate and is dominated by pine forests. No cases of scrub typhus have been reported in the past decades. Yulong county has a low-latitude highland South Asian monsoon climate. Only a few cases of scrub typhus have been reported in Yulong county in the past decades, and the incidence peaked in 2015 at 9.10 per 100,000. No studies on *Ot* infection of small mammals were reported in the three counties. However, the three counties are a potential natural reservoir for vector-borne disease because the diverse climates and rich vegetation benefit the survival and reproduction of small mammals and ectoparasite populations, contributing to potentially neglected concerns for the residents and travelers.

The study was conducted throughout a calendar year during all four seasons, including winter (which started from 30 November to 31 December 2015), spring (6 March to 29 March 2016), summer (7 July to 16 July 2016), and autumn (9 October to 18 October 2016). Lianghe was divided into four elevation gradients (1000–1200 m, 1200–1400 m, 1400–1600 m, and more than 1600 m). Jianchuan was divided into three elevation gradients (2250–2450 m, 2450–2650 m, and more than 2650 m). Yulong was divided into four elevation gradients (2400–2600 m, 2600–2800 m, 2800–3000 m, and more than 3000 m).

### 2.2. Small Mammals Collection and DNA Extraction

Small mammals were captured by snap traps (15 × 8 cm, Metal rat trap clip, China), and the species were identified by key morphology characteristics of small mammals. Spleen or liver tissue from each specimen was collected using aseptic techniques and stored at −40 ℃. Relevant environmental parameters (elevation, landscape) were recorded during the fieldwork (Trimble Juno SB, Los Altos, CA, USA), and the details of small mammal capture and tissue collection were previously described [19]. Deoxyribonucleic acid (DNA) from the spleen and liver was extracted by the BioTeke Whole blood genomic DNA Kit (AU19014-16, BioTeke Corporation, Beijing, China), according to the manufacturer’s instruction and then stored at −40 ℃ until molecular assays could be conducted.

### 2.3. NPCR Amplification and Sequencing

Concentration and purification of extracted DNA were initially assessed using NanoDrop 2000/2000 c Spectrophotometers (ND-2000, Thermo Fisher Scientific, Waltham, MA, USA). When the extract concentration was ≥50 μg/mL, and A_260_/A_280_ was between 1.8 to 2.0, it was selected as the detection template [20]. Nested Polymerase Chain Reaction (NPCR) was performed to amplify the gene sequence based on the 56 kDa outer membrane protein of *Ot*, two sets of primers used were from the technical guidelines for preventing and controlling scrub typhus (2009) from the Chinese center for disease control and prevention, China (China CDC) described and listed in Table 1. The primers were synthesized, and the primer concentration was 10 μM (Sangon Biotech, Shanghai, China).

The NPCR was carried out with 25 μL reaction mixtures. The first round of reaction contained 1 μL each of outer primers a and b, 12.5 μL of Dream Taq Green PCR Master Mix (K1081, Thermo Fisher Scientific, Waltham, MA, USA), and 5 μL of the template. The NPCR conditions consisted of pre-denaturation at 94 °C for 5 min, denaturation at 94 °C for 1 min, annealing at 50 °C for 1 min, and extension at 72 °C for 1 min, followed by 35 cycles. Finally, extension at 72 °C for 5 min. The second round of reaction contained 1 μL each of inner primers a’ and b’, 1μL of the first-round amplification product was produced as the template, and the other reaction systems and conditions were the same as the first round. Blank and negative controls were included throughout the entire process. The second-round products were collected and electrophoresed in 1.5% agarose gel containing Gelview (BioTeke Corporation, Beijing, China) and visualized under a Gel imaging system (G: BOX F3, Syngene, Frederick, MD, USA). The amplified products between 150–168 bp were confirmed as *Ot* positive and then sequenced in both directions for additional verification (Sangon Biotech, Shanghai, China).

### 2.4. Phylogenetic Analysis

Successfully detected sequences were edited and trimmed by DNASTAR (version 7.1). Reference complete or partial sequences encoding 56-kDa TSA of *Ot* were retrieved from GenBank by the Blast program of the National Center for Biotechnology Information (https://blast.ncbi.nlm.nih.gov/Blast.cgi, 13 January 2019). Sample sequences were aligned with reference sequences using Sequence distance in Meglign of DNASTAR. Phylogenetic analysis was performed with the Clustal W protocol (default parameters) by Mega software (version 7.0). Phylogenetic trees were constructed by a neighbor-joining method after 1000 bootstrapped replicates [21,22]. In addition, the phylogenetic trees were also constructed by a maximum likelihood method to confirm the validity. The reference strains from the study are listed in Appendix A.

### 2.5. Statistical Analysis

Demographic, geographic, and laboratory parameters were recorded in Microsoft Excel 2016. Wild small mammals are classified as dominant species (>10%) and other species (≤10%) according to the constituent ratio. Based on the amount of DNA successfully extracted from wild small mammals and *Ot* detection results, the *Ot* infection rate in wild small mammals was calculated by wild small mammal samples infected with *Ot* as the numerator, and the number of wild small mammals DNA successfully extracted as the denominator.

The *Ot* infection in different species, sex, county, season, and elevation was compared by Chi-square Test or Fisher’s Exact Test using R software (4.0.2 version). *P* values less than 0.05 were considered statistical significance.

## 3. Results

### 3.1. Species of Wild Small Mammals and Ot Infection

A total of 2538 wild small mammal specimens were collected from the three counties, including 22 species of Rodentia, 7 species of Insectivora, and 1 species of Scandentia. *Apodemus chevrieri* were the dominant species, which accounted for 30.89% (784/2538), followed by *Eothenomys miletus* 19.11% (485/2538) and *Apodemus draco* 10.87% (276/2538) (Table 2). In Lianghe, a total of 630 small mammals were identified; *Rattus tanezumi* (28.89%, 182/630) was the dominant species, followed by *Rattus sladeni* (19.37%, 122/630) and *Mus pahari* (11.59%, 73/630). In Jianchuan, the dominant species were *A. chevrieri* (46.46%, 479/1031), *E. miletus* (32.78%, 338/1031), and *A. draco* (11.45%, 118/1031). In Yulong, the dominant species were *A. chevrieri* (34.78%, 305/877), *A. draco* (18.02%, 158/877), *E. miletus* (15.96%, 140/877), and *Apodemus latronum* (11.86%, 104/877).

Out of 2538 samples, 45 tested positive based on the 56 kDa TSA gene with an overall prevalence of 1.77% (Table 2). Thirteen of thirty species tested positive, as demonstrated by the agarose gel electrophoresis results of NPCR from the second amplified products (150–168 bp), shown in Figure 2. The *Ot* infection rate varied from 0.26 to 9.09% by different species. The prevalence of *Ot* was highest in *Niviventer andersoni* (9.09%, 1/11), but the sample size was too small. The second prevalence was in *Rattus rattus* (7.32%, 9/123), and the lowest was in *A. chevrieri* (0.26%, 2/784). The infection rate of *Ot* in wild small mammals from Lianghe, Jiangchuan, and Yulong counties was 2.85%, 2.13%, and 0.57%, respectively. A total of 18 samples, including six species, were positive in Lianghe; the prevalence of *Ot* was highest in *R. rattus* (7.38%, 9/122) and was lowest in *Hylomys suillus* (1.96%, 1/51). Of the 22 samples with five species positive for *Ot* in Jianchuan, 15 *E. miletus* were positive, but there were no positives in the other two counties. In Yulong, only *A. draco*, *A. latronum,* and *Niviventer confucianus* were positive for *Ot*, and *N. confucianus* had the highest *Ot* infection rate (2.27%, 1/44) (Table 2).

### 3.2. Phylogenetic Analysis Based on the 56-kDa TSA Gene of Ot

Out of 45 sample sequences (18 from Lianghe, 22 from Jianchuan, and the remaining 5 from Yulong county) (Appendix A), the 56 kDa TSA gene nucleotide homology and phylogenetic analysis was conducted, and 13 of the sample sequences aligned with 18 reference sequences (Figure 3), which demonstrated 56.6–100% identity to *Ot* strains. Among the 45 sample sequences, 35.6% (16/45) shared 91.6-97% identity with *Ot Gilliam* strains (DQ485289 et al.), and 11.1% (5/45) of sequences shared 80.7–99.5% identity with *Ot Karp* strains (M33004 et al.). One sequence shared 92.9% identity with *Ot TA763* strains (MK660519 et al.). In addition, the sample sequences of JC-4-91, SL-3-132, SL-4-006, SL-4-18, YL-3-288, YL-4-94, YL-4-102, and YL-4-121 (eight sample sequences) were similar and shared 81.1–100% identity with each other. They also shared 81.8–86.2% identity with the Thailand strain (HM777460) and 80.5–86.8% identity with the Taiwan of China strain (GU446605). The sequences of JC-1-74, JC-1-106, JC-2-78, JC-3-66, SL-1-34, SL-3-56, SL-3-97, SL-3-115, SL-4-009, SL-4-19, SL-4-47, SL-4-106, YL-2-190, LH-3-18, and LH-3-162 (15 sample sequences) were similar and shared 83.6–100% identity with each other, and among them, the sequence of JC-1-74, JC-1-106, JC-2-78, JC-3-66, SL-1-34, SL-3-56, SL-3-97, SL-3-115, SL-4-009, SL-4-19, SL-4-47, SL-4-106, YL-2-190, and LH-3-162 shared 90.1–96.1% identity with the Yunnan strain (KU664516). In contrast, LH-3-18 shared 93.4% identity with the Taiwan of China strain (GQ332755) but only shared 87.5% identity with the Yunnan strain (KU664516).

The phylogenetic tree constructed with 56 kDa TSA gene sequence homologies of 13 sample sequences and 18 reference sequences of *Ot* is shown in Figure 4. The phylogenetic trees constructed using the maximum method demonstrate a similar distribution (Appendix A). The sample sequences of LH-3-167, LH-3-171, LH-3-24, JC-3-13, and SL-3-114 close to the *Gilliam* strain (DQ485289 et al.) were identified as *related-Gilliam* genotypes. The sample sequences of LH-3-63, LH-4-183, and LH-2-110 close to the *Karp* strain (M33004 et al.) were identified as *related-Karp* genotypes. LH-2-102 had a higher identity with the Thailand TA763 strain, which was identified as *related-TA763* strain. YL-4-94 and JC-4-94 were close to the Taiwan (GU446605) and Thailand strains (HM777460), which could be identified as close to the *Karp* genotype. In addition, SL-3-97, SL-4-19, YL-2-190, and LH-3-18 sequences, although close to the Yunnan (KU664516) and Taiwan strains (GQ332755), resolve independently in one branch.

### 3.3. Associated Environmental and Seasonal Factors for Ot Infection

When comparing *Ot* infection in wild small mammals from the three counties sampled in this study, the infection prevalence in wild small mammals from Lianghe county was higher than that in Jianchuan and Yulong counties. Comparing the prevalence of *E. miletus* between the different species of wild small mammals, the prevalence was highest in the three dominant species (*A. chevrieri*, *E. miletus*, *A. draco*), followed by *A. draco* and *A. chevrieri* (*p* < 0.05) (Table 3). However, there were no significant differences (*p* > 0.05) in the season, landscape, and sex of small mammals for the *Ot* infection in wild small mammals.

Regarding wild small mammals captured at various elevations, the prevalence of wild small mammals in 1200–1400 m was higher than the other three elevations (*p* = 0.009) in Lianghe county (Table 4). In Yulong county, the infection prevalence in wild small mammals from 2800–3000 m was higher among the four elevations (*p* < 0.05). However, there was no significant difference (*p* > 0.05) from 2250 to 2850 m in Jianchuan county.

## 4. Discussion

Scrub typhus is a remerging vector-borne disease and has become one of the most prevalent diseases in Asia-Pacific, including China. The incidence of scrub typhus in humans is influenced by climate change and the distribution of small mammals with parasitic mites. Therefore, infected small mammals contribute to the local spread of scrub typhus [23]. In Yunnan, scrub typhus cases were primarily reported from May to December, but few studies on *Ot* infection in small mammals were reported. In this study, 2538 wild small mammals were captured seasonally from Lianghe, Jianchuan, and Yulong counties. The overall prevalence of *Ot* was 1.77%, and *Gilliam*, *Karp*, and *TA763* were the main genotypes of *Ot*. The infection rate was consistent with previous studies, and the prevalence ranged from 0.4–23.08% in small mammals in Yunnan [16,18,24]. This prevalence was slightly lower than detected in Shandong (2.6%) and Anhui (8.3%), located in East and Central China, respectively [14,25]. The prevalence of *Ot* varies according to the species of small mammals captured, the captured period of small mammals, and the habitat and elevation of small mammals captured. Moreover, the low infection rate of *Ot* could result from the unfavorable climate and high elevation in Yulong county, which are not beneficial for the maintenance and transmission of *Ot* in small mammals.

Among the sampled wild small mammals, the dominant species of wild small mammals from each county was consistent with previous studies, which listed *R. tanezumi* as the main species of small mammals in the commensal plague natural foci (Lianghe county), *A. chevrieri* and *E. miletus* were the dominant species of small mammals in wild rodents plague natural foci (Jianchuan and Yulong counties)in Yunnan province [26,27]. Moreover, *Mus Pahari*, *A. draco*, and *A. latronum* were reported as the new dominant species of wild small mammals in Lianghe, Jianchuan, and Yulong counties, respectively, indicating that recent increases in human activities, agricultural land use, and the decrease in primary forests have led to changes in small mammal abundance and reproduction, which indirectly affects the wild small mammals’ community composition. However, more research is needed to determine whether the small changes in the distribution component of the dominant wild small mammals in the three counties are stable over time or coincidental. The species of small wild mammals determine the species and number of parasitic mites carried, and the composition of dominant small mammals in an area also indirectly affects the on-host abundance of infected mites and influences the transmission of *Ot* in the natural environment. Therefore, understanding the species and distribution of dominant wild small mammals can also contribute to effective modeling towards understanding the transmission of *Ot* in nature, resulting in disease prevention for humans.

Small mammals are the main host for mites, and they carry infected mites long distances. Therefore, the distribution and species of small mammals are of practical importance to understanding the natural transmission of *Ot*. This knowledge will aid in the prediction of possible maintenance of scrub typhus in nature and the risk of humans acquiring *Ot* from nature [28]. Previous studies demonstrated that several small mammal species are capable of being infected with *Ot*; however, the infection rate varies. In this study, 13 small mammal species can become infected with *Ot* and show the potential risk for transmitting *Ot* in nature. *R. tanezumi*, *N. confucianus,* and *N. andersoni* are known reservoir hosts of *Ot* in Yunnan province. In this study, the same three species were infected with *Ot*. Moreover, *A. draco*, *A. latronum*, *N. fulvescens*, *Suncus murinus*, *Anourosorex squamipes*, and *Crocidura attenuate* were also infected with *Ot*. Although these survey areas did not include the region where the natural foci of scrub typhus are located, there are diverse species of wild small mammals infected with *Ot*. This could indicate the potential for the presence of a scrub typhus natural focus in western Yunnan. The infection rate of *Ot* in *E. miletus* was 3.09%, which was consistent with previous studies reporting that *E. miletus* was infected with *Ot* in Yunnan province [18]. Previous studies have mostly focused on the eastern and central areas of Yunnan province [16,18]. However, in western Yunnan, water and dense vegetation are more abundant, resulting in beneficial climates for mite survival. These include tropical and subtropical climates with diverse small mammal and vector populations.

*Ot* is genetically unique among other rickettsia that have been previously studied. This uniqueness is partially based on the 56 kDa TSA gene or groEL gene observed from samples collected from humans or animals in some provinces of China, such as Shandong, Jiangsu, Jiangxi Anhui, and Yunnan [13,14,24,29]. In this study, the analysis revealed that *Ot* sequences were identified between 56.6–100% identical to the reference *Ot* strain sequences from humans, mites, and small mammals. The *Gilliam* genotype was the most common strain at a rate of 35.6%, followed by the *Karp* and *TA763* strains. This result was consistent within Guangdong, which detected that the *Karp*, *Gilliam,* and *TA763* strains could be identified in small mammals, but differs from the results reported from Korea, which showed that the *Boryong* strain (85.7%) was the most common strain in small mammals [6,30]. It is known that *Karp*, *Gilliam*, and *Kato* were the predominant strains of *Ot* in Yunnan [18]. However, in this study, the *TA763* strain was also identified as a common strain in Lianghe county in addition to the *Karp* and *Gilliam* strains. While Ya et al. found that *Kawasaki* strains existed in Yunnan [18], our study found that some sample strains were closer to *Kawasaki* strains but derived independently in one branch of our phylogenies. This result could potentially be explained by the shorter sequence or mutations in some parts of the amplified products of positive samples. Further investigation on *Kawasaki* strains from this study is warranted.

The *Ot* infection rate first rose and then fell, accompanied by an increase in elevation. In Lianghe county, the *Ot* infection rate at 1200–1600 m was significantly higher than that in 1000–1200 m. In Yulong county, the *Ot* infection rate at 2800–3000 m was also significantly higher than that in more than 3000 m. Different populations of small mammals respond differently in beneficial environments. This indirectly affects the population of parasitic mites and the resulting transmission of *Ot*. Fewer individuals and mammal species were collected at an elevation of more than 3000 m, which may be the result of a harsher environment resulting in a lack of water and vegetation.

It has been shown that *Ot* is seasonally transmitted in humans or small mammals. In humans from Korea, scrub typhus increased in October and peaked in November. Alternatively, human cases of scrub typhus in Japan are consistent throughout the year, but differences are observed in the northern parts of the island compared to the south [31]. In China, scrub typhus is most often diagnosed in humans during the summer-autumn seasons in the southern focus area but during the autumn-winter seasons in the north focus area [32,33]. In this study, *Ot* was detected in all four seasons from Jianchuan county. However, in Lianghe and Yulong counties, *Ot* detection only occurred during the summer, autumn, and winter. No significant difference in small mammal infection rates for *Ot* exists when all four seasons are compared. However, more infected small mammals were detected during the autumn-winter season, suggesting that additional infected mites could be found in nature, resulting in an increased risk of potential *Ot* infection. This study captured small wild mammals primarily in the forest, followed by farmland and scrub, which likely influenced infection rates. It cannot be ignored that habitat is one of the critical conditions for the long-term survival of small mammals and mites, indirectly affecting *Ot* transmission and overall reservoir capacity. Different habitat conditions support different small mammal populations, which each have their own *Ot* reservoir ability. *Ot*-infected small mammals were identified from multiple habitats, suggesting that infected small mammals or infected mites can transport and spread *Ot* within an ecosystem. Additional research is needed to better understand how this pathogen persists and expands its distribution.

Some limitations of this study include only sampling three counties in western Yunnan and not including a study site within the natural focus of scrub typhus. However, these three counties were chosen because they have abundant small mammals, according to long-time monitors for the institute of plague control and prevention department. This abundance contributes to the potential risk for all rodent-borne diseases. Because of the diverse assemblage of small mammals and infection rates similar to those observed in the scrub typhus foci areas, our results indicate that these areas are at risk of becoming foci. Also, small mammals captured during this study were not limited to the accepted main *Ot* hosts, such as *R. tanezumi* and *R. rattus*. Because of this broad sampling, this study observed more than 10 species of small mammals infected with *Ot*, confirming the diversity of *Ot* infection in small mammals. More research within these areas is warranted to further examine important hosts for *Ot*. Finally, *Ot* was detected based on the 56 kDa TSA gene, but the sample sequences were too short to distinguish mutations in some samples. The 56 kDa TSA gene is a dominant protein of *Ot*. It has been a diverse antigen and includes variants such as *Karp*, *Gilliam*, and *Kato*, among others [34].

## 5. Conclusions

A diverse assemblage of small mammal species collected from western Yunnan was infected with *Ot*. *Rattus rattus* is the main reservoir host with *Ot*, while *Karp*, *Gilliam,* and *TA763* strains were the main genotypes detected in the small mammal tested. To our knowledge, this is the first report of the TA763 strain detected in western Yunnan. People living in Lianghe and Jianchuan counties may have a higher risk for *Ot* infection, especially when *Ot* spills over from nature. This study confirms that infection rates were highest in autumn and winter; thus, extra precautions should be taken to avoid human infection. Elevations between 1200–1400 m in Lianghe and 2800–3000 m in Yulong county had a higher risk for infected *Ot* in small mammals. These findings suggest that further monitoring of environmental conditions and their impact on *Ot* infection in diverse small mammals is required. Additional surveillance will likely provide adequate information on the existence of novel scrub typhus focus in western Yunnan.

## Figures and Tables

**Figure 1 pathogens-12-00128-f001:**
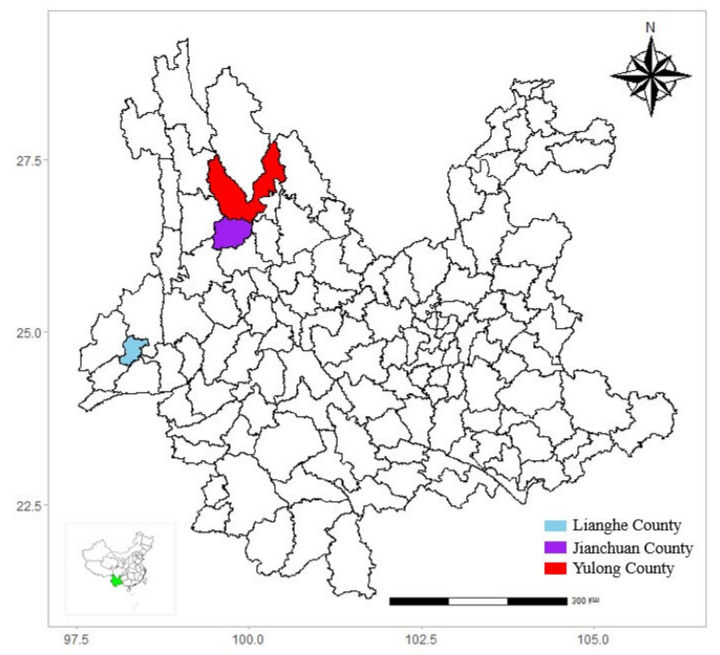
Counties sampled for wild small mammals in western Yunnan Province, China.

**Figure 2 pathogens-12-00128-f002:**
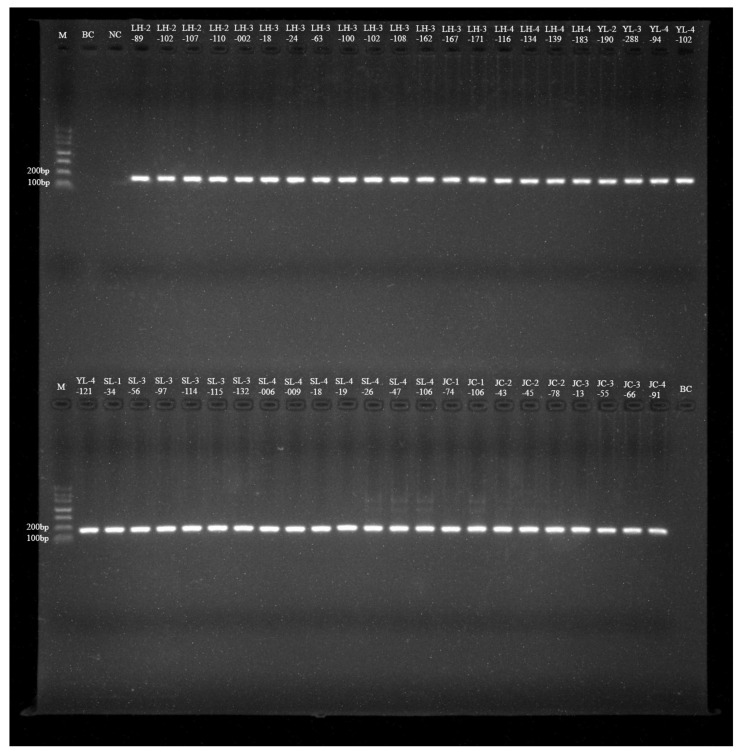
Agarose gel electrophoresis results of NPCR from *Ot*-infected wild small mammals in the three counties of western Yunnan Province. M: DNA Marker Ⅰ, from 700 to 100 bp; BC: Black control, NC: Negative control; LH-2-89 to JC-4-91: NPCR positive product.

**Figure 3 pathogens-12-00128-f003:**
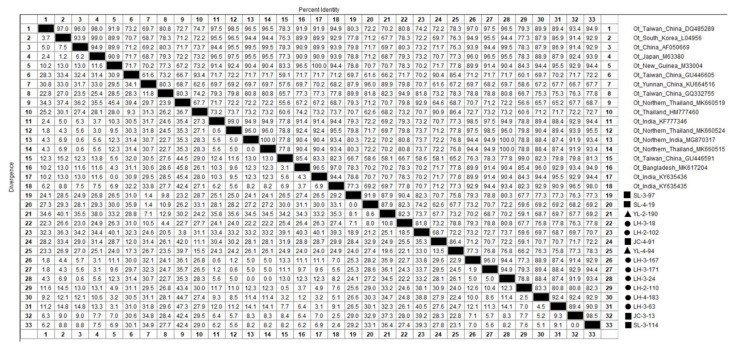
The 56 kDa TSA nucleotide homology matrix of Ot strain and reference strain in western Yunnan Province. 
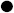
 Positive samples from Lianghe; 
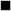
 Positive samples from Jianchuan; 
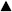
 Positive samples from Yulong.

**Figure 4 pathogens-12-00128-f004:**
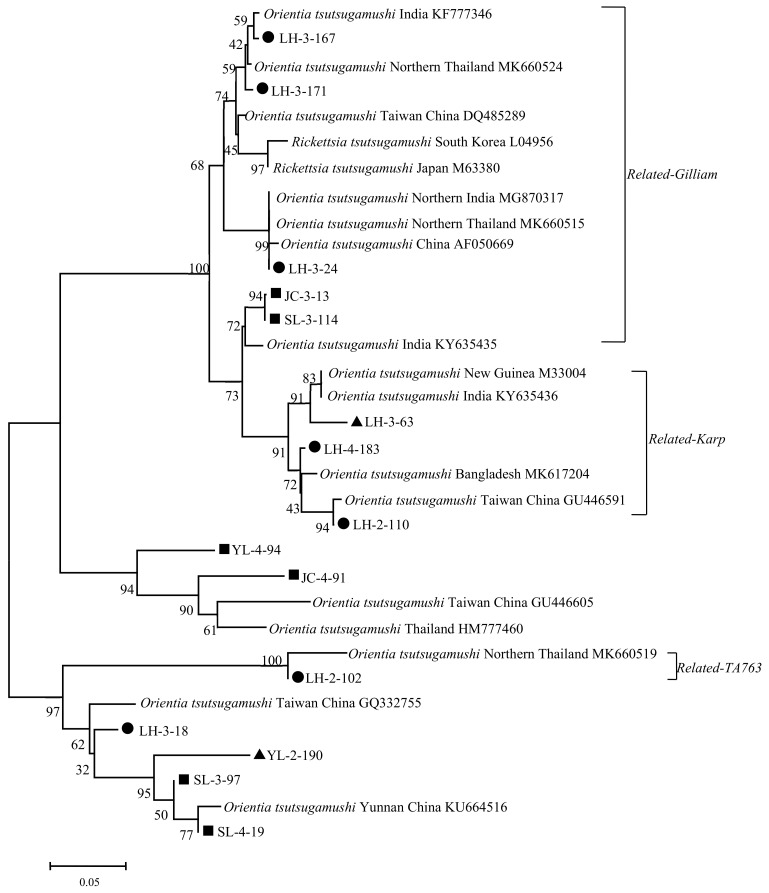
Phylogenetic tree based on Ot 56 kDa gene fragment. 
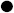
 Positive Samples from Lianghe; 
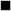
 Positive Samples from Jianchuan; 
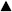
 Positive Samples from Yulong.

**Table 1 pathogens-12-00128-t001:** The detection primers of *Ot* based on the 56 kDa TSA gene in NPCR.

Primers	Primer Sequence	Amplified Fragments
Out Primer		
a	5’- TACATTAGCTGCGGGTATGACA-3’	306~339 bp
b	5’- CCAGCATAATTCTTCAACCAAG-3’	
Inner Primer		
a’	5’- GAGCAGAGCTAGGTGTTATGTA-3’	150~168 bp
b’	5’- TAGGCATTATAGTAGGCTGAGG-3’	

**Table 2 pathogens-12-00128-t002:** Distribution of *Ot* infections on wild small mammals collected from the three counties of western Yunnan Province * [positive/*n* (%)].

Species	Lianghe County	Jianchuan County	Yulong County	Total
*Apodemus chevrieri*	0	2/479 (0.42)	0/305	2/784 (0.26)
*Apodemus draco*	0	3/118 (2.54)	3/158 (1.90)	6/276 (2.17)
*Apodemus latronum*	0	0/22	1/104 (0.96)	1/126 (0.79)
*Rattus nitidus*	0/12	0	0/1	0/13
*Rattus norvegicus*	0	0/4	0	0/4
*Rattus rattus/sladeni*	9/122 (7.38)	0/1	0	9/123 (7.32)
*Rattus tanezumi*	5/182 (2.75)	0/2	0/2	5/186 (2.69)
*Niviventer confucianus*	0/2	0/9	1/44 (2.27)	1/55 (1.82)
*Niviventer fulvescens*	1/62 (1.61)	0	0	1/62 (1.61)
*Niviventer andersoni*	0/2	1/6 (16.67)	0/3	1/11 (9.09)
*Micromys minutes*	0	0/6	0	0/6
*Mus pahari*	0/73	0	0	0/73
*Vernaya fulva*	0	0	0/1	0/1
*Bandicota indica*	0/1	0	0	0/1
*Berylmys bowersi*	0/9	0	0	0/9
*Eothenomys miletus*	0/7	15/338 (4.44)	0/140	15/485 (3.09)
*Eothenomys proditor*	0	0	0/56	0/56
*Eothenomys Eleusis*	0/17	0	0	0/17
*Eothenomys melanogaster*	0	0/2	0	0/2
*Dremomys pernyi*	0	0/1	0/20	0/21
*Sciurotamias forresti*	0	0	0/1	0/1
*Collosciurus erythraeus*	0	0	0/1	0/1
*Suncus murinus*	1/46 (2.17)	3	0	1/49 (2.04)
*Sorex minutes*	0	0	0/2	0/2
*Crocidura dracula*	0/3	0/3	0/6	0/12
*Soriculus leucops*	0	0	0/4	0/4
*Anourosorex squamipes*	1/31 (3.23)	0/1	0	1/32 (3.13)
*Crocidura attenuate*	0/6	1/12 (8.33)	0/6	1/24 (4.17)
*Hylomys suillus*	1/51 (1.96)	0	0	1/51 (1.96)
*Tupaia belangeri*	0/4	0/24	0/23	0/51
Total	18/630 (2.86)	22/1031 (2.13)	5/877 (0.57)	45/2538 (1.77)

* Alphabetically arranged according to genus.

**Table 3 pathogens-12-00128-t003:** Factor analysis of *Ot* infection rates in wild small mammals collected from the three counties of western Yunnan Province.

Factors	Sample	Positive Sample	Infection Rate (%)	*p*
**County**				<0.05
Lianghe	630	18	2.86	
Jianchuan	1031	22	2.13	
Yulong	877	5	0.57	
**Season**				0.057
Spring	537	3	0.56	
Summer	548	8	1.46	
Autumn	806	19	2.36	
Winter	647	15	2.32	
**Landscape**				0.802 *
Woodland	1661	27	1.63	
Cultivation	341	7	2.05	
Scrub	70	1	1.43	
Cultivation & Woodland	273	5	1.83	
Cultivation & Scrub	193	5	2.59	
**Species**				<0.01 *
*A. chevrieri*	784	2	0.26	
*E. miletus*	485	15	3.09	
*A. draco*	276	6	2.17	
Other	993	22	2.22	
**Sex**				0.906
Male	1169	20	1.71	
Female	1354	24	1.77	

* Notes: Fisher’s Exact Test. Other *p* values were from the Chi-square Test.

**Table 4 pathogens-12-00128-t004:** Comparison of the *Ot* infection in wild small mammals in different elevations of each county.

County	Elevation (m)	Sample	Positive Sample	Infection Rate (%)	*p*
Lianghe					0.009 *
	1000~1200	175	0	0	
	1200~1400	206	10	4.85	
	1400~1600	156	6	3.85	
	≥1600	93	2	2.15	
Jiangchuan					0.822 *
	2250~2450	154	2	1.3	
	2450~2650	646	15	2.32	
	≥2650	231	5	2.16	
Yulong					0.013 *
	2400~2600	81	1	1.23	
	2600~2800	115	1	0.87	
	2800~3000	169	3	1.78	
	≥3000	512	0	0	

* Notes: Fisher’s Exact Test. Other *p* values were from the Chi-square Test.

## Data Availability

The datasets used and/or analyzed during the current study are available from the first and corresponding author upon request.

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
