# Peer review of "Orientia tsutsugamushi Infection in Wild Small Mammals in Western Yunnan Province, China"

_pathogens, 2023, doi:10.3390/pathogens12010128_

Round 1
Reviewer 1 Report
The article is sound and worthy of publication, however, some areas within the text were difficult to understand. I went through each line and attempted to revise the text to more clearly explain what I thought you were writing. Please review the attached document and revise to better explain each section, especially the discussion.

Reviewer 2 Report
Fundamentally a solid study. The below suggestions are important points that I believe require clarification.
Suggestions:
1) Title "Orientia tsutsugamushi infection on wild small mammals in 2 western Yunnan Province, China"- I would change to "in wild small animals" as reads better
2) How did you calculate your sample size?
3) No reference to vivisection qualifications of the investigators?
4) Why did it take so long to publish results from 2016/17?
Round 2
Reviewer 1 Report
Please see additional edits in attached file

Author Response
Dear reviewer,
The authors thank the reviewers for their revisions and agree with the changes. The author modified (red color) and added the comment "The second revised" in the text. Please check it.
At the same time, the author has slightly different opinions in line"281"(revised version in lines "302-303"): A. chevrieri and E. miletus were the dominant species of small mammals in wild rodents plague natural foci (Jianchuan and Yulong counties)in Yunnan province[26, 27]. The authors think keeping the "in wild rodents plague natural foci" because in Yunnan Province there are two plague natural foci, including wild rodents plague natural foci and the commensal rodents plague natural foci. Jianchuan and Yulong counties belong to wild rodents plague natural foci.
Best regards,
The authors
.

Reviewer 2 Report
Would still change title to in, not on small mammals.
Satisfied with responses otherwise.
Author Response
Dear reviewer,
Thanks for your suggestion, the authors also agree with you, and the author has changed the title to "Orientia tsutsugamushi infection in wild small mammals in western Yunnan Province, China" in the revised MS.
Best regards,
The authors